# Proteomic Analysis Reveals Key Proteins in Extracellular Vesicles Cargo Associated with Idiopathic Pulmonary Fibrosis In Vitro

**DOI:** 10.3390/biomedicines9081058

**Published:** 2021-08-20

**Authors:** Juan Manuel Velázquez-Enríquez, Jovito Cesar Santos-Álvarez, Alma Aurora Ramírez-Hernández, Edilburga Reyes-Jiménez, Armando López-Martínez, Socorro Pina-Canseco, Sergio Roberto Aguilar-Ruiz, María de los Ángeles Romero-Tlalolini, Luis Castro-Sánchez, Jaime Arellanes-Robledo, Verónica Rocío Vásquez-Garzón, Rafael Baltiérrez-Hoyos

**Affiliations:** 1Facultad de Medicina y Cirugía, Universidad Autónoma Benito Juárez de Oaxaca, Oaxaca de Juárez 68120, Oaxaca, Mexico; juanmanuelvela_enriquez@live.com (J.M.V.-E.); jovitocesarsa@hotmail.com (J.C.S.-Á.); aramih_09@hotmail.com (A.A.R.-H.); edilreyesjimnez@yahoo.com.mx (E.R.-J.); armandoloopez37@gmail.com (A.L.-M.); sar_cinvestav@hotmail.com (S.R.A.-R.); 2Centro de Investigación, Facultad de Medicina, Universidad Nacional Autónoma de México-Universidad Autónoma Benito Juárez de Oaxaca, Oaxaca de Juárez 68120, Oaxaca, Mexico; socopina12@hotmail.com; 3CONACYT-Facultad de Medicina y Cirugía, Universidad Autónoma Benito Juárez de Oaxaca, Oaxaca de Juárez 68120, Oaxaca, Mexico; mdlaromerotl@conacyt.mx (M.d.l.Á.R.-T.); veronicavasgar@gmail.com (V.R.V.-G.); 4CONACYT-Universidad de Colima, Centro Universitario de Investigaciones Biomédicas “CUIB”, Universidad de colima, Av. 25 de Julio No. 965, Col. Villas San Sebastián, Colima 28045, Colima, Mexico; luis_castro@ucol.mx; 5CONACYT-Instituto Nacional de Medicina Genómica, 14610 Ciudad de México, Mexico; jarellanes@inmegen.gob.mx

**Keywords:** idiopathic pulmonary fibrosis, fibroblasts, extracellular vesicles, proteomic analysis, mass spectrometry

## Abstract

Idiopathic pulmonary fibrosis (IPF) is a chronic, progressive, irreversible, and highly fatal disease. It is characterized by the increased activation of both fibroblast and myofibroblast that results in excessive extracellular matrix (ECM) deposition. Extracellular vesicles (EVs) have been described as key mediators of intercellular communication in various pathologies. However, the role of EVs in the development of IPF remains poorly understood. This study aimed to characterize the differentially expressed proteins contained within EVs cargo derived from the fibroblast cell lines LL97A (IPF-1) and LL29 (IPF-2) isolated from lungs bearing IPF as compared to those derived from the fibroblast cell lines CCD8Lu (NL-1) and CCD19Lu (NL-2) isolated from healthy donors. Isolated EVs were subjected to label-free quantitative proteomic analysis by LC-MS/MS, and as a result, 331 proteins were identified. Differentially expressed proteins were obtained after the pairwise comparison, including all experimental groups. A total of 86 differentially expressed proteins were identified in either one or more comparison groups. Of note, proteins involved in fibrogenic processes, such as tenascin-c (TNC), insulin-like-growth-factor-binding protein 7 (IGFBP7), fibrillin-1 (FBN1), alpha-2 collagen chain (I) (COL1A2), alpha-1 collagen chain (I) (COL1A1), and lysyl oxidase homolog 1 (LOXL1), were identified in EVs cargo isolated from IPF cell lines. Additionally, KEGG pathway enrichment analysis revealed that differentially expressed proteins participate in focal adhesion, PI3K-Akt, and ECM–receptor interaction signaling pathways. In conclusion, our findings reveal that proteins contained within EVs cargo might play key roles during IPF pathogenesis.

## 1. Introduction

Idiopathic pulmonary fibrosis (IPF) is a chronic, progressive, and irreversible disease of somewhat uncertain etiology and pathogenesis [1,2]. IPF is characterized by increased fibroblasts and myofibroblasts proliferation that results in exacerbated production and accumulation of extracellular matrix (ECM), promoting pulmonary parenchyma remodeling, which in turn leads to pulmonary insufficiency, and eventually, patient death [2]. Although its etiology and pathogenesis have not been fully described, it has been proposed that the IPF onset is stimulated by chronic insults on the lung epithelium caused by exposure to environmental agents such as tobacco smoke, metal, and silica dust, as well as by some microbial agents [3,4]. In addition, some genetic variants have also been strongly associated with IPF development [3,4]. For example, variants of the *MUC5B* gene, encoding for mucin 5B protein, are associated with the maintenance of bronchoalveolar epithelial function [3,4,5]. Likewise, variants of *TERT* and *TERC* genes, which encode for telomerase reverse transcriptase, and the RNA component of telomerase, respectively, associated with the telomere length maintenance, telomere shortening is a feature that has been associated with the development of IPF [3,4].

IPF preferentially appears in the sixth decade of life and is more frequent in men who have smoked for a long time [6]. Once IPF is diagnosed, its prognosis has a survival of 3–4 years [2,6]. Its high mortality is largely due to the lack of knowledge underlying the molecular mechanisms leading the disease development [7]. Therefore, the identification of key molecules involved in the pathophysiological processes that promote the IPF progression remains an unmet need [7,8]. Of note, a variety of investigations have proposed that extracellular vesicles (EVs) play a pivotal role as mediators of cell–cell communication, as well as they have highlighted the role of EVs as cooperators in the development of lung diseases, including IPF [8,9,10]. EVs are small membrane vesicles secreted by a wide variety of cells in both physiological and pathological conditions, and they are classified according to their size as either exosomes (40–150 nm) or microvesicles (50–1000 nm) [9,10]. Interestingly, EVs carry a wide variety of bioactive molecules such as DNA, RNA, lipids, and proteins, which confer them the capability to regulate the activity and behavior of the host cell by activating different signaling pathways [8,10].

In recent decades, proteomic analysis has become a valuable tool for the large-scale characterization of proteins expression in different chronic diseases. Thus, this tool has both facilitated and hastened the identification of key proteins involved in the progression of different diseases [7,11]. Coincidently, several investigations on IPF have used proteomic approaches to characterize the molecular mechanisms involved in the disease progression [2,3,7,12,13,14,15,16]; however, limited studies have aimed to identify the proteomic profile of EVs cargo secreted during IPF progression [8]. Thus, here we identified differentially expressed proteins contained within EVs cargo secreted by cell lines isolated from lungs bearing IPF.

## 2. Materials and Methods

### 2.1. Cell Lines and Culture Conditions

The fibroblast cell lines LL97A and LL29, bearing a human IPF phenotype, were purchased from the American Type Culture Collection (ATCC, Cat. No. CCL-134 and CCL-191, respectively; Manassas, VA, USA). The fibroblast cell lines CCD8Lu and CCD19Lu, used as normal human lung fibroblasts, were purchased from the American Type Culture Collection (ATCC, Cat. No. CCL-201 CCL-210, respectively; Manassas, VA, USA). All cell lines were cultured in Eagle’s minimal essential medium (EMEM) (ATCC No. 30-2003) supplemented with 10% SFB and 100 U/mL penicillin/streptomycin (GIBCO). Cells were maintained in a humidified incubation at 37 °C and 5% CO_2_, and experiments were performed using passages from 8 to 12.

### 2.2. Isolation of Extracellular Vesicles and Protein Extraction

EVs were obtained from culture supernatants of CCD8Lu (NL-1), CCD19Lu (NL-2), LL97A (IPF-1), and LL29 (IPF-2) cell lines using ultracentrifugation. Briefly, cell lines were grown in 10 cm culture plates to 70% confluence. Then, cells were washed three times with PBS before adding fresh culture medium supplemented with 5% SFB EXO-DEPLETED to reduce contamination of EVs derived from SFB and other nanoparticles that may interfere with the further analysis. After incubation for 48 h (85–95% confluence), culture supernatants were collected and differentially centrifuged at 2000× *g* for 30 min at 4 °C to remove cell debris, followed by 12,000× *g* for 30 min at 4 °C to remove cell debris and apoptotic bodies. Supernatants were recovered and centrifuged at 100,000× *g* for 3 h at 4 °C in a TH641 rotor; then, EVs pellets were resuspended in PBS, carefully washed, and centrifuged at 100,000× *g* for 3 h at 4 °C. EVs were resuspended in CHAPS lysis buffer for protein extraction, and protein concentration was measured by Bradford’s method. Finally, to validate that EVs were selectively isolated, specific EVs markers were detected in isolated proteins by Western blot. In addition, for label-free quantitative proteomic analysis, isolated EVs were resuspended in 8 M urea lysis buffer. Protein concentration was measured by bicinchoninic acid (BCA) assay (Thermo Fisher Scientific, Waltham, MA, USA).

### 2.3. Western Blot

Western blot (WB) analysis was performed according to standard protocols. Briefly, 50 µg of total protein isolated from EVs were separated by SDS-PAGE gel electrophoresis. Then, proteins were transferred to a PVDF membrane and primary antibodies Anti-HSP90-αβ (1:500; SC13119; Santa Cruz Biotechnology, Dallas, TX, USA), Anti-Alix (1:500; SC53540; Santa Cruz Biotechnology), and Anti-Flot-1 (1:500; BD 610821; BD Biosciences, Franklin Lakes, NJ, USA) were incubated overnight at 4 °C. Protein expression was visualized after membranes were incubated with horseradish peroxidase-conjugated anti-mouse secondary antibodies (1:5000; SC-516102; Santa Cruz Biotechnology), and then protein spots were revealed by using 1-Step Ultra TMB-Blotting reagent (Thermo Fisher Scientific, Waltham, MA, USA).

### 2.4. Label-Free Quantitative Proteomic Analysis

#### 2.4.1. Chemicals and Instrumentation

DL-dithiothreitol (DTT), iodoacetamide (IAA), formic acid (FA), acetonitrile (ACN), and methanol were purchased from Sigma (St. Louis, MO, USA), and bovine pancreatic trypsin was purchased from Promega (Madison, WI, USA). Ultrapure water was prepared from a Millipore purification system (Billerica, MA, USA). An Ultimate 3000 nano UHPLC system along with the Q Exactive HF mass spectrometer (Thermo Fisher Scientific, Waltham, MA, USA) with an ESI nanospray source were used.

#### 2.4.2. Sample Preparation

First, total protein lysates from EVs samples were centrifuged at 16,000× *g*, 4 °C for 15 min, and supernatants were transferred to a clean Eppendorf tube. Then, total proteins were precipitated from protein solution by using cold acetone and centrifuged at 16,000× *g*, 4 °C for 15 min, and supernatant was discarded. Subsequently, protein pellets were dissolved in 6 M of aqueous urea solution, and 30 μg of total protein was denatured with 10 mM DTT by incubating it at 56 °C for 1 h, followed by alkylation with 50 mM IAA and incubated at room temperature for 60 min in the dark. Next, 500 mM ammonium bicarbonate was added to the solution to obtain a final concentration of 50 mM ammonium bicarbonate at pH 7.8. Then, trypsin (Promega) was added to protein solution at 1:50 ratio for digestion at 37 °C for 15 h. Digested peptides were further purified with a zip tip to remove the salt. Finally, samples were dried under vacuum and stored at −20 °C for further analysis.

#### 2.4.3. Nano LC-MS/MS Data Collection

LC-MS/MS data were collected using an Ultimate 3000 nano UHPLC system coupled to a Q Exactive HF mass spectrometer (Thermo Fisher Scientific, Waltham, MA, USA) and an ESI nanospray source. Four comparative groups were analyzed, meaning a total of 8 samples, i.e., two biological replicates per group.

#### 2.4.4. Nano-LC

Reversed-phase nano-LC separation was performed on the Ultimate 3000 nano UHPLC system (Thermo Fisher Scientific, Waltham, MA, USA). A total of 1 μg of sample from collected fractions was loaded onto a capture column (PepMap C18, 100 Å, 100 μm × 2 cm, 5 μm), and then, samples were separated on an analytical column (PepMap C18, 100 Å, 75 μm × 50 cm, 2 μm). The mobile phases used were (A) 0.1% FA in water and (B) 0.1% FA in 80% ACN. A linear gradient was applied from 2% to 8% buffer B for 3 min, from 8% to 20% buffer B for 56 min, from 20% to 40% buffer B for 37 min, and then, from 40% to 90% buffer B for 4 min at 250 nL/min flow rate.

#### 2.4.5. Mass Spectrometry

The full scan was performed between 300 and 1650 *m*/*z* with a resolution of 60,000–200 *m*/*z*, and the automatic gain control target for full scan was set at 3.0 × 10^6^ MS/MS scan was operated in Top 20 mode using the following settings: resolution 15,000 at 200 *m*/*z*; automatic gain control target 1.0 × 10^5^ maximum injection time 19 ms; collision energy was normalized at 28%; 1.4 Th isolation window; charge state exclusion: unassigned, 1 > 6; and dynamic exclusion was 30 s.

#### 2.4.6. Analysis of LC-MS/MS Data

The raw MS files were analyzed and searched against the human protein database based on sample species using Maxquant (v1.6.2.6) (Max Planck Institute, Martinsried, Germany). Parameters were set as follows: protein modifications were carbamidomethylation (C) (fixed), oxidation (M) (variable); enzyme specificity was set to trypsin; maximum missed cleavages were set to 2; precursor ion mass tolerance was set to 10 ppm, and MS/MS tolerance was 0.6 Da. The false discovery rate (FDR) of peptides and proteins was 0.01. Proteins were quantified using LFQ (Label-Free Quantification) intensity. After processing in Maxquant, proteins without MS/MS and LFQ intensity were removed. Proteins with LFQ ≠ 0 in at least 1 out of 8 samples (2 biological replicates per EVs group) were retained for further analysis using the Perseus v1.6.15.0 (Max Planck Institute, Martinsried, Germany) bioinformatics platform [17]. Then, LFQ intensities were log2 transformed to achieve a normal data distribution. Results were filtered to remove potential contaminants, reverse matches, and proteins only identified by site. Subsequently, they were filtered for retaining proteins with a MS/MS spectral count ≥2; then, data were row filtered according to valid values (minimum valid percentage, 75%) and then normalized by median subtraction. Further imputation of missing values was performed by selecting a downward shift of 1.8 and a width of 0.3 standard deviations in a normal distribution. Fold change (FC) was calculated by subtracting the average of log2 values (Δlog2 (LFQ intensity)) between proteins identified in IPF-1 and IPF-2 vs. proteins identified in NL-1 and NL-2 groups. The comparison was performed in four pairwise groups, i.e., IPF-1 vs. NL-1, IPF-1 vs. NL-2, IPF-2 vs. NL-1, IPF-2 vs. NL-2, by using Student’s *t*-test for two-tailed unpaired data. *p*-values were adjusted using FDR-based permutation method; therefore, a corrected *p*-value of 0.05 with an FC ≥ 1 and ≤−1 was set as the cutoff for determining whether a protein was differentially expressed. Protein identification data were further processed and visualized using the statistical software GraphPad Prism 8 (San Diego, CA, USA) and RStudio (Boston, MA, USA). Heat maps were generated using Morpheus platform (https://software.broadinstitute.org/morpheus, accessed on 2 June 2021).

### 2.5. Computational Annotations and Bioinformatics

A 4-way Venn diagram was performed for identified proteins in each EVs sample using the interactiVenn Web application (http://www.interactivenn.net/, accessed on 22 May 2021) [18]. Gene ontology (GO), biological process (BP), molecular function (MF), cellular component (CC), and KEGG pathway enrichment annotation analysis of differentially expressed proteins were performed online using the gene annotation co-occurrence discovery (GeneCodis) classification system (https://genecodis.genyo.es/, accessed on 29 May 2021) [19,20,21]. GO and KEGG pathway analyses were considered significant when *p*-values were <0.05. In addition, to evaluate the interaction and functional enrichment between differentially expressed proteins, the search tool for retrieval interacting genes/proteins v11.5 (STRING) was used (https://string-db.org/, accessed on 30 May 2021) [22,23]. Cytoscape v3.8.2 software (Cytoscape Consortium, San Diego, CA, USA) was used to visualize the network, and we also used cluster analysis through Molecular Complex Detection Complement (MCODE) for the identification of core proteins [24].

### 2.6. Statistical Analysis

Student’s *t*-test for unpaired, two-tailed data was used to determine the statistical significance of FC in log2-transformed proteomic data.

## 3. Results

### 3.1. Isolation and Characterization of Extracellular Vesicles

Supernatants were purified from NL-1, NL-2, IPF-1, and IPF-2 cultured cell lines by differential centrifugation (Figure 1A). Western blot analysis from EVs-isolated proteins showed that extracts were highly enriched in Alix, HSP90, and Flotilin-1, three well-known EVs markers (Figure 1B), indicating that samples obtained from the cell culture supernatants were enriched with proteins contained within EVs cargo.

.

### 3.2. Overview of Proteomic Analysis and Reproducibility of LC-MS/MS Data

Purified proteins were individually obtained from each EVs sample and digested with trypsin. LC-MS/MS data were acquired on a Q Exactive HF mass spectrometer using two biological replicates per cell line type. The raw MS files were analyzed and searched in UniProt human protein database using the Maxquant platform. Proteins recording an LFQ intensity value ≠ 0 in at least one out of eight total samples (two biological replicates per group) were considered as identified. A total of 567 proteins were identified in all isolated EVs (Appendix A). Then, identified proteins were filtered to remove those showing an MS/MS spectral count of either 0 or 1; as a result, a total of 331 proteins were obtained (Appendix A), which were considered for further analysis. Figure 2A shows the experimental workflow followed for proteomic analysis, as explained in detail in the Methods section.

LFQ intensity was used to determine the relative abundance of proteins in each sample. Consistency in LFQ intensity values proves to be paramount to obtain an accurate measurement of protein abundance identified in samples. Box plot for log2 values of LFQ intensity per sample, including their respective biological replicate, indicates that interquartile range and median are similar between duplicates in each sample. This result indicates that LC-MS/MS measurements across replicates were consistent (Figure 2B). In addition, we also obtained an adequate determination coefficient (R2) of LFQ intensities per sample as well as their biological replicates (Figure 2C). R2 values between replicates for EVs proteome were: 0.97 for NL-1 and NL-2, 0.92 for IPF-1, and 0.99 for IPF-2. Obtained data suggest a higher correlation between the replicates in each sample than between those in different EVs samples.

### 3.3. Identification of Differentially Expressed Proteins

LFQ intensity expression values and MS/MS spectral counts were used for classifying the 331 proteins into the four EVs groups. Proteins were considered specifics of a group only when the LFQ intensity value was different from zero, and MS/MS was either equal to two or higher, at least in one out of two biological replicates. The Venn diagram shows both the common and specific proteins present in the proteome of each EVs sample derived from different cell lines. (Figure 3A). A total of 262 proteins were identified in NL-1, 292 in NL-2, 273 in IPF-1, and 274 in IPF-2; of which 215 proteins were common in all EVs groups. The number of specific proteins was 9 for NL-1, 11 for NL-2, 10 for IPF-1, and 6 for IPF-2 (Appendix A).

Three proteins were identified exclusively from IPF-2 and IPF-1 groups, namely, lysyl oxidase homolog 1 (LOXL1), collagen and calcium-binding EGF domain-containing protein 1 (CCBE1), and fibulin-2 (FBLN2). In addition, results that showed 10 proteins were exclusively for NL-1 and NL-2 groups, namely, collagen alpha-1 (VIII) chain (COL8A1), prostaglandin F2 receptor negative regulator (PTGFRN), complement component C8 beta chain (C8B), basigin (BSG), integrin alpha-5 (ITGA5), dihydropyrimidinase-related protein (DPYSL2), integrin alpha-1 (ITGA1), dihydropyrimidinase (DPYS), CD109 antigen (CD109), and SH3 domain-binding glutamic acid-rich-like protein 3 (SH3BGRL3).

LFQ intensity values were used to calculate the number of differentially expressed proteins. Protein FC was calculated by subtracting the average of log2 values (Δlog2 (LFQ intensity)) between the proteins identified from IPF-1 and IPF-2 vs. proteins identified from NL-1 and NL-2, and the comparison was performed in four pairwise groups; i.e., IPF-1 vs. NL-1, IPF-1 vs. NL-2, IPF-2 vs. NL-1, IPF-2 vs. NL-2). Proteins showing an FC ≥ 1 but ≤−21 and *p*-value < 0.05 were considered significant.

Figure 3B shows differentially expressed proteins in the four comparisons. Results indicate that 7 proteins from IPF-1 and 38 proteins from IPF-2 (≥1 and *p* < 0.05) were up-regulated, as well as 10 proteins from IPF-1 and 20 proteins from IPF-2 were down-regulated (≤−1 and *p* < 0.05) as compared to the proteome of NL-1 (Appendix A). Moreover, results also show that 5 proteins from IPF-1 were up-regulated and 15 proteins from IPF-2 (≥1 and *p* < 0.05), as well as 14 proteins from IPF-1 were down-regulated and 13 proteins from IPF-2 (≤−1 and *p* < 0.05) as compared to the proteome of NL-2 (Appendix A).

Volcano plots show differentially expressed proteins between IPF-1 vs. NL-1 (Figure 3C), IPF-1 vs. NL-2 (Figure 3D), IPF-2 vs. NL-1 (Figure 3E), and IPF-2 vs. NL-2 (Figure 3F). Additionally, differentially expressed proteins were clustered and visualized in a heat map showing a consistent expression pattern and clustering between groups (Appendix A). Differentially expressed proteins identified in two or more comparison groups were 22, of which 13 were up-regulated, and 9 were down-regulated (Appendix A).

### 3.4. Functional Ontology Classification and Pathway Enrichment Analysis of Differentially Expressed Proteins

Proteins identified as expressed differentially from IPF-1 and IPF-2 vs. those in NL-1 and NL-2 were included for GO annotation analysis. The analysis revealed that proteins associated with IPF were mainly involved in BP, MF, and CC processes. Proteins associated with BP, such as tenascin-c (TNC), insulin-like-growth-factor-binding proteins 7 (IGFBP7), and fibrillin-1 (FBN1), were up-regulated and enriched for extracellular matrix organization and cell adhesion (Figure 4A). Among down-regulated proteins, annexin A2 (ANXA2), annexin A1 (ANXA1), and aminopeptidase N (ANPEP) were associated with neutrophil degranulation and drug response (Figure 4A).

From proteins associated with the MF process, up-regulated proteins were mainly involved in extracellular matrix structural constituent and proteins binding, including TNC, IGFBP7, and FBN1 (Figure 4B). On the other hand, down-regulated proteins were mainly involved in protein binding and identical proteins binding, including collagen alpha-1(XVIII) chain (COL18A1), annexin A5 (ANXA5), and 5-nucleotidase (NT5E) (Figure 4B).

From proteins associated with CC, up-regulated proteins were mainly involved in the collagen-containing extracellular matrix and the extracellular region, including TNC, IGFBP7, and FBN1 (Figure 4C). Down-regulated proteins were mainly involved in the extracellular exosome and extracellular region, including COL18A1, ANXA5, and NT5E (Figure 4C). The full list of GO analyses can be found in Appendix A.

In addition, KEGG pathway enrichment analysis showed that up-regulated proteins were involved in ECM-receptor interaction, focal adhesion, and PI3K-Akt signaling pathways. Proteins related to these pathways were laminin subunit beta-2 (LAMB2), laminin subunit alpha-2 (LAMA2), laminin subunit alpha-5 (LAMA5), and TNC (Figure 5). Down-regulated proteins were mainly enriched in focal adhesion, PI3K-Akt signaling, protein adsorption, and protein digestion pathways. Proteins related to these pathways were COL18A1, filamin-C (FLNC), and integrin alpha-2 (ITGA2) (Figure 5). The full list of KEGG pathway enrichment analysis can be found in Appendix A.

### 3.5. Protein Network Analysis

Enrichment analysis of the KEGG pathway showed that some differentially expressed proteins belong mainly to focal adhesion, PI3K-Akt, and ECM-receptor interaction pathways. These pathways have been described to be enriched during IPF development [25,26,27,28]. To gain a better understanding about of the role of these proteins in IPF, all differentially expressed proteins from four comparisons were used to construct an interaction network using STRING v11.5 software and Cytoscape v3.8.2. Results show that differentially expressed proteins form a complex interaction network containing 86 nodes and 239 edges with an average node degree of 5.56 and a clustering coefficient of 0.615. The expected number of edges was 46, which was much lower than the actual edges encountered, and the PPI enrichment *p*-value was <1.0 × 10^−16^ (Figure 6A). Thus, our results indicate that this network has key biologically connected interactions that might impact the behavior of IPF progression.

In addition, the entire network was analyzed using the MCODE plug-in from Cytoscape software. A, significant module was identified with an average MCODE score = 10.6, nodes = 21, and edges = 106. This module comprised 21 differentially expressed proteins; including TNC, IGFBP7, FBN1, collagen alpha-2 (V) chain (COL5A2), collagen alpha-1 (V) chain (COL5A1), collagen alpha-1 (III) chain (COL3A1), alpha-2 collagen chain (I) (COL1A2), alpha-1 collagen chain (I) (COL1A1), alpha-1 collagen chain (VI) (COL6A1), COL18A1, collagen alpha-1 (IV) chain (COL4A1), collagen alpha-1 (XV) chain (COL15A1), serpin H1 (SERPINH1), LAMB2, metalloproteinase inhibitor 1 (TIMP1), insulin-like-growth-factor-binding protein 3 (IGFBP3), insulin-like-growth-factor-binding protein 5 (IGFBP5), sulfhydryl oxidase 1 (QSOX1), stanniocalcin-2 (STC2), follistatin-related protein 1 (FSTL1), and versican (VCAN) (Figure 6B).

## 4. Discussion

IPF is characterized by an increased agglomeration of fibroblasts bearing a profibrotic phenotype [29]. These cells have been described as the main source of the extracellular matrix that accumulates in fibrotic areas of the lung [4,29]. Several investigations have shown that lung fibroblasts isolated from IPF patients and cultured in vitro conserve a phenotypic profile that maintains a high capability to express markers such as smooth muscle alpha actin (α-SMA), extracellular matrix-associated proteins collagen I or fibronectin, as well as a higher proliferative capability compared to fibroblasts isolated from normal lungs [29,30,31].

In this study, a label-free quantitative proteomic analysis was used to identify differentially expressed proteins in EVs cargo derived from the fibroblast cell lines LL97A (IPF-1) and LL29 (IPF-2), and compared with those EVs cargo derived from the fibroblast cell lines CCD8Lu (NL-1) and CCD19Lu (NL-2). Three proteins were identified to be exclusive of IPF-1 and IPF-2, namely, LOXL1, CCBE1, and FBLN2. Moreover, a total of 17 differentially expressed proteins were identified in IPF-1 vs. NL-1, 58 in IPF-2 vs. NL-1, 19 in IPF-1 vs. NL-2, and 28 in IPF-2 vs. NL-2 comparison. KEGG pathway enrichment analysis showed that a significant number of differentially expressed proteins in EVs cargo derived from fibroblasts bearing IPF phenotype were mainly related to focal adhesion, PI3K-Akt, and ECM-receptor interaction signaling pathways, which demonstrates that proteins captained in EVs cargo from an IPF phenotype might favor the pathogenesis of IPF.

The key role of EVs in intercellular communication has been mainly associated with the molecular cargo, such as proteins with capabilities to activate signaling pathways that in turn regulate the activity and behavior of the host cell [8,9]. Previous studies have used proteomic approaches to uncover the cell-derived EVs proteins involved in the pathogenesis of IPF [8]. For example, elevated levels of fibronectin have been found on the surface of fibroblast-derived EVs, showing a senescent phenotype; additionally, it has been shown that fibronectin-enriched EVs promote an invasive phenotype on recipient fibroblasts by interacting with α5β1 integrin and, in turn, stimulate the activation of signaling pathways that involve FAK and Src, two kinases involved in cell invasion [8].

Therefore, identification of differentially expressed proteins in EVs cargo secreted by cultured IPF lung fibroblasts may provide relevant data on the main pathophysiological mechanisms associated with the role of fibroblasts in the IPF progression.

It is important to point out that some of the proteins identified as exclusive of IPF-1 and IPF-2 groups have been studied because of their close relationship with the IPF pathogenesis. For example, LOXL1 is an enzyme that belongs to the lysyl oxidase (LOX) family and catalyzes the cross-linking between collagen and elastin by promoting the oxidative deamination of lysine residues and contributing to the stiffness of ECM in different tissues [32,33,34]. Moreover, it has been shown that the main LOXL1-producing cells are fibroblasts and myofibroblasts in the fibrotic lung, and this enzyme has been mainly localized into dense fibrosis and fibroblastic foci [33,34]. Increased LOX1 levels have been associated with increased proliferation, invasion, resistance to apoptosis, and shrinkage of fibroblasts [33,34]. Studies in murine models of IPF have demonstrated that LOXL1-/- mice are protected from IPF induced by AdTGF-β1 administration [32,34]. However, the role of LOXL1 in the pathogenesis of IPF has not been fully described.

FBLN2 is a glycoprotein belonging to the fibulin family, which is predominantly localized in the basement membrane of the extracellular matrix and elastic basal fibers [35]. It has been described that FBLN2 can regulate different signaling pathways associated with cell metabolism, migration, proliferation and differentiation [35]. FBLN2 is also up-regulated during the tissue remodeling process, and its overexpression has been associated with different chronic diseases such as fibrosis and cancer [35,36]. In vitro studies performed in cardiac fibroblasts isolated from Fbln2-/- mice showed that FBLN2 plays an essential role in angiotensin II-induced TGF-β signaling pathway, while studies performed in a murine model of cardiac hypertrophy demonstrated that Fbln2-/- mice show attenuated development of cardiac hypertrophy induced by chronic infusion of subpressor and pressor doses of angiotensin II [36]. Additionally, the role of FBLN2 has been also studied in a murine model of myocardial fibrosis induced by chronic infusion of pressor doses of angiotensin II showing that the fibrosis degree decreases in Fbln2-/- mice [37]. However, its participation in the pathogenesis of IPF had not been described.

The involvement of CCBE1 in IPF has not been described. However, CCBE1 has been reported to be associated with extracellular matrix remodeling, development of multicellular organisms and development of lymphatic vessels [38,39]. Currently, the relationship of CCBE1 with different types of cancer has been studied [39]. However, data indicate that CCBE1 may have particular effects depending on the type of cancer, which has generated some controversy about its role in cancer development [38]. An investigation showed that in patients bearing lung cancer, CCBE1 expression is down-regulated within tumor tissue which is associated with poor prognosis [39]. In contrast, CCBE1 is overexpressed in colorectal cancer tissue compared to adjacent tissues, Results showed that CCBE1 overexpression significantly correlates with increased lymph node metastasis, vascular invasion, and liver metastasis [38]. Thus, our result indicates that CCBE1 might be also participating in the IPF progression and suggests that it could be contributing in the fibrogenesis associated with cancer, such as the fibrosis that precede the liver cancer appearance, an intriguingly phenomenon that should be addressed.

Interestingly, some up-regulated proteins that we identified had already been reported as key players in the pathogenesis of fibrotic diseases, such as TNC, IGFBP7, FBN1, COL5A2, COL5A1, COL3A1, COL1A2, COL1A1 and COL6A1) [40,41,42,43]. Interestingly, these proteins were identified as part of a relevant module in the PPI network of differentially expressed proteins, indicating that these proteins are biologically interconnected.

Some of these proteins are involved in IPF development through different mechanisms. For example, TNC is an EMC-associated hexameric glycoprotein that is transiently expressed during tissue injury [43,44,45]. Additionally, it has been described that TNC regulates fibroblast migration and cell adhesion [43,44]. In lung tissue from IPF patients, TNC is up-regulated as compared to lung tissue from healthy patients [44]. Furthermore, studies in a bleomycin-induced murine model of IPF have demonstrated that TNC-/- mice exhibit significant fibrosis attenuation [43]; indicating that TNC plays an important role in fibrogenesis processes such as IPF and suggesting that this role might be closely associated with its release as part of the EVs cargo.

IGFBP7, a glycoprotein that belongs to the IGFBP superfamily, has been little studied in IPF. However, this protein has been shown to promote various biological processes associated with fibrotic disorders in different organs; for example, it is involved in the development of kidney and liver fibrosis [41,46]. In this context, stimulation of tubular epithelial cells by TGF-β enhances IGFBP7 expression, and that its silencing decreases the number of α-SMA-positive cells induced by TGF-β [46]. Similarly, it has been evidenced that IGFBP7 promotes hepatic stellate cells (HSC) activation, favors the development of a myofibroblast phenotype, and promotes ECM protein production [41]. Furthermore, a significant increase in IGFBP7 expression has been reported in lung tissue of IPF patients [47]. Taken together, our results suggest that IGFBP7 protein contributes to the pathogenesis of IPF and that its transport within EVs cargo might be an altered mechanism essential for its functioning.

To our knowledge, the functional role of FBN1 has not been described in IPF. However, the involvement of FBN1 in the development of different fibrotic disorders has already been reported [48,49]. For example, it has been revealed that microfibrils isolated from the skin of Tsk-/- mice, a model of systemic sclerosis, are enriched in FBN1 and promote a prooxidant phenotype in cultured endothelial cells which contributes to the mesenchymal transition [48]. In addition, FBN1 is overexpressed in the liver, undergoing a fibrogenic process induced by carbon tetrachloride [49].

Additionally, the stimulation of rat HSC and human hepatic myofibroblasts with TGF-β was shown to promote FBN1 production. [49]. On the other hand, it has been demonstrated that FBN1 is overexpressed in renal tissue and that its deletion decreases the injury and fibrosis induced by unilateral ischemia-reperfusion injury (UIRI) in a murine model of renal fibrosis. In addition, FBN1 has been described to inhibit proliferation and promote endothelial cell apoptosis [42]. However, to our knowledge, it had not been reported yet whether FBN1 contained in EVs plays a role in the development of other fibrotic diseases such as IPF. Thus, our result encourages further clarifying this intriguing proposal.

Interestingly, our results show that several collagens family members were up-regulated in IPF, including COL5A2, COL5A1, COL3A1, COL1A2, COL1A1, and COL6A1. The collagen family consisting of 28 different collagen subtypes represents the major component of the airway ECM [50]. They also play an important function in providing the ECM an increased tensile strength [50,51]. It has been described that collagen plays a particular role in the pathogenesis of IPF [50]. Increased collagen deposition in the lung alveolar walls results in progressive destruction of normal alveolar architecture, resulting in increased ECM stiffness [50]. Under normal and fibrotic conditions, type I collagen is the main component in the airway ECM, consisting of two α-1 and one α-2 chain encoded by the *COL1A1* and *COL1A2* genes, respectively [51].

Several studies have reported the important role of type I collagen in the development of IPF, as well as its regulation by TGF-β [51]. Collagen type I has previously been described as an inducer of mesenchymal cell differentiation into myofibroblasts characterized by an increased α-SMA expression [52]. Our results contrast with previous investigations about the expression and secretion of different collagen chains in fibroblasts derived from the lungs of IPF patients [53]. In this regard, it has been described that upon fibroblasts stimulation with TGF-β, type I collagen (COL1A1 and COL1A2), type III collagen (COL3A1), and type V collagen (COL5A1 and COL5A2) are up-regulated [53]. However, to date, there is no evidence on the biological role that different collagen, carried within EVs cargo, play in the development of IPF.

In the present study, we identified differentially expressed proteins in EVs cargo isolated from human IPF fibroblast cell lines that may play a key role in the progression of IPF. The results shown in this study may provide new insights into the molecular mechanisms of IPF. However, some limitations of this investigation, such as the need for additional experimental evidence that can validate the involvement of the identified proteins either in in vivo systems or in samples from patients bearing IPF, have yet to be addressed. Moreover, the association between the content of EVs cargo and IPF must also be investigated.

## 5. Conclusions

In conclusion, by using the label-free quantitative proteomic technology, we analyzed the content of EVs cargo isolated from cell lines bearing an IPF phenotype and identified differentially expressed proteins associated with human IPF progressions, such as TNC, IGFBP7, FBN1, COL1A1, COL1A2, and LOXL1. Therefore, proteins contained within EVs cargo might be playing key roles during IPF progression. Additionally, further studies are still needed to conclusively clarify the role of proteins transported within EVs cargo in the IPF pathogenesis.

## Figures and Tables

**Figure 1 biomedicines-09-01058-f001:**
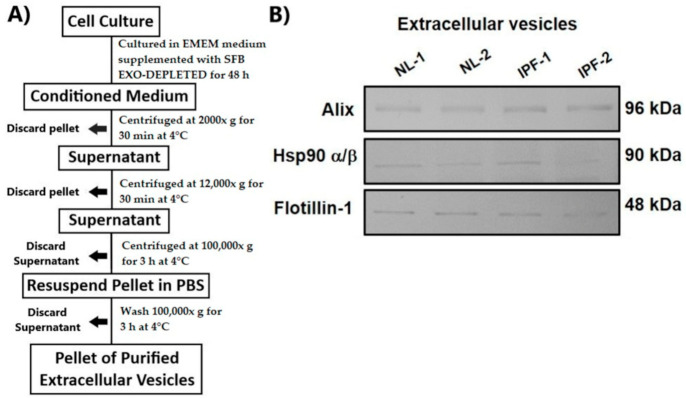
Isolation and characterization of EVs. (**A**) EVs isolation protocol. (**B**) Western blot analysis of EVs markers Alix (96 kDa), Hsp90 α/β (90 kDa), and Flotilin-1 (48 kDa). IPF, idiopathic pulmonary fibrosis; NL, normal lung.

**Figure 2 biomedicines-09-01058-f002:**
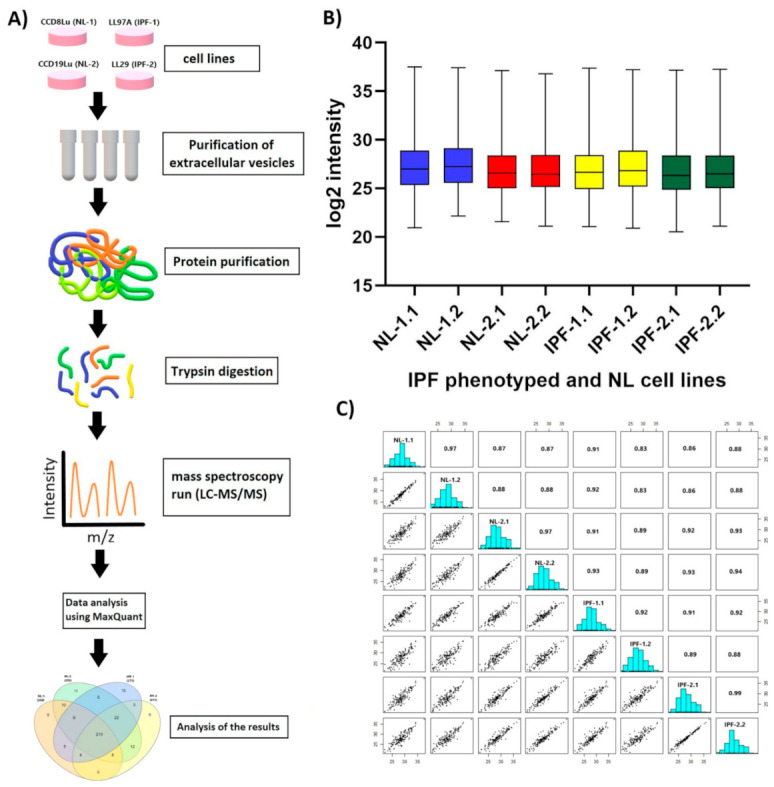
Experimental design of proteomic analyses and reproducibility of LC-MS/MS data. (**A**) Experimental workflow of proteomic analyses. (**B**) Box plot of biological replicates of each EVs sample obtained from each cell line. (**C**) Correlation plot of biological replicates per EVs sample obtained from each cell lines. IPF, idiopathic pulmonary fibrosis; NL, normal lung.

**Figure 3 biomedicines-09-01058-f003:**
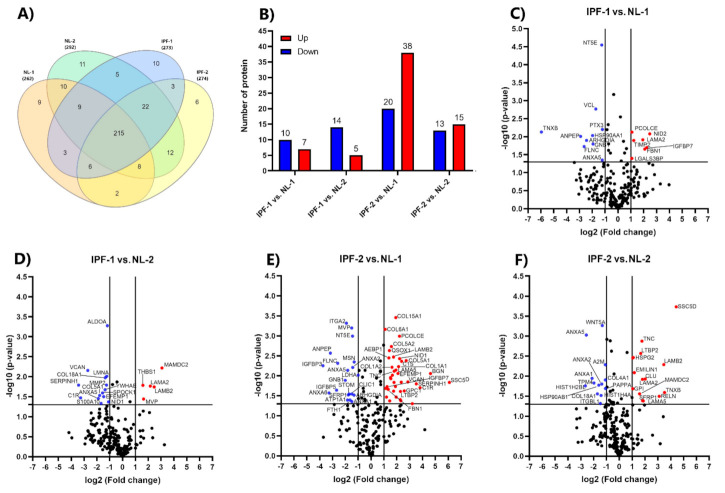
Differentially expressed proteins. (**A**) Venn diagram shows protein distribution identified by LC-MS/MS in comparative groups. (**B**) Total up-regulated (red) and down-regulated (blue) proteins identified in all pairwise comparisons. (**C**–**F**) Volcano plot showing both up-regulated (red) down-regulated (blue) proteins in all pairwise comparisons. Volcano plots depict FC (*x*-axis) and −log10 value of *p*-value (*y*-axis). Red dots in the upper right (ratio ≥1) and blue dots in the upper left (ratio ≤−1) sections represent significantly deregulated proteins, *p* < 0.05. IPF, idiopathic pulmonary fibrosis; NL, normal lung.

**Figure 4 biomedicines-09-01058-f004:**
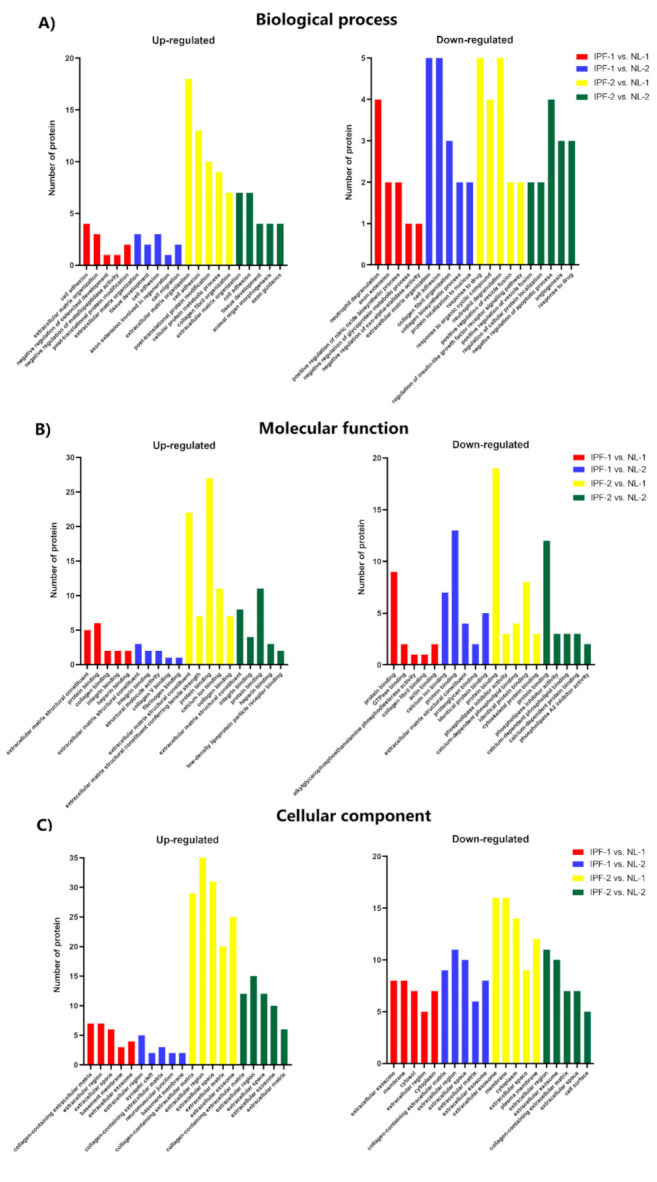
GO analysis showing differentially expressed proteins in five main processes. (**A**) Up- and down-regulated proteins in biological processes. (**B**) Up- and down-regulated proteins in molecular functions. (**C**) Up- and down-regulated proteins in the major cellular components. IPF, idiopathic pulmonary fibrosis; NL, normal lung.

**Figure 5 biomedicines-09-01058-f005:**
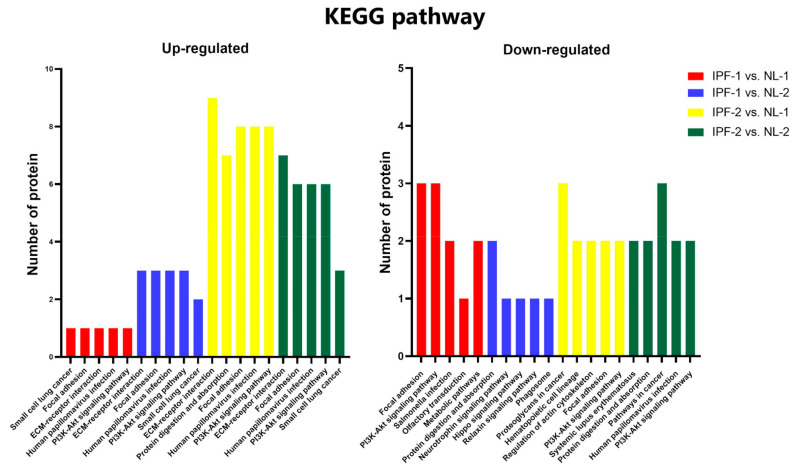
KEGG pathway enrichment analysis of differentially expressed proteins. Up- and down-regulated proteins are involved in the top five KEGG pathways. IPF, idiopathic pulmonary fibrosis; NL, normal lung.

**Figure 6 biomedicines-09-01058-f006:**
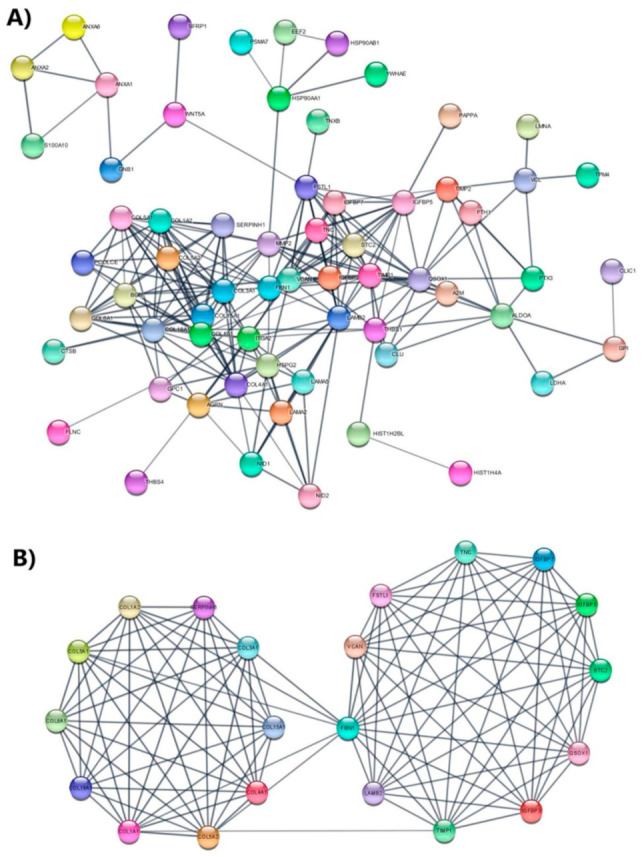
Protein–protein interaction regulatory network of differentially expressed proteins. (**A**) Differentially expressed proteins were combined to construct a regulatory network using both STRING and Cytoscape software to visualize the interaction and functional enrichment with evidence as network edge significance and active interaction sources were Text Extraction, Experiments, Database, Coexpression, Neighborhood, Gene Fusion, and Co-occurrence, with a minimum required interaction score being high confidence (0.7). (**B**) Significant module identified from PPI network through Cytoscape MCODE complement; degree cutoff = 2, node score cutoff = 0.2, k-core = 2 and maximum depth = 100.

## Data Availability

Data presented in this study are available on request from the corresponding author.

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
