# Peer review of "Proteomic Analysis Reveals Key Proteins in Extracellular Vesicles Cargo Associated with Idiopathic Pulmonary Fibrosis In Vitro"

_biomedicines, 2021, doi:10.3390/biomedicines9081058_

Round 1

Reviewer 1 Report

In this manuscript, Velázquez et al. describe a set of experiments in which they extract vesicles from 2 healthy lung-derived and 2 IPF lung derived fibroblast lines, subject them to label free relative quantification by LC/MS/MS, and further characterize these differential protein profiles using bioinformatic methods.

The paper has merit and presents some interesting data but, in my opinion, is hampered by a few major weaknesses that should be addressed.

A notable weakness of this study, and of any other study that only investigates fibroblasts cultured in vitro, is the non-physiological culture conditions that fibroblasts are exposed to when cultured in vitro. In vitro, fibroblasts maintain some characteristics that they possessed in vivo, while losing others, limiting consequences that can be extrapolated solely from in vitro findings. The findings that the authors present here would be enhanced significantly if they demonstrated that the IPF-derived fibroblasts (preferably cultured under the conditions that the authors used to produce vesicles) demonstrated increased pro-fibrotic phenotypes compared to the healthy lung fibroblasts (increased collagen deposition, increased expression of pro-fibrotic genes like ACTA2, COL1A1, TGFB1, CCN2, etc., by IF, WB, RT-PCR, etc.), or other relevant phenotypes that would suggest that some IPF phenotypes in vivo are maintained in fibroblasts cultured in vitro relative to "healthy" lung fibroblasts. If the authors haven't demonstrated that these fibroblasts maintain some differential, IPF-related phenotypes, then it will be very difficult to convince the reader that the differential protein expression profiles that they find in their proteomic analyses (and that they use for downstream pathway analyses, etc.) are relevant to fibrosis in some way. This weakness is also exacerbated by the differences in culture medium and FBS concentration that the authors describe in the methods for culture of IPF vs. non-IPF fibroblasts. 

The weaknesses of conclusions solely from in vitro study should be stated and expanded upon in the discussion section by the authors, in order to put their study in greater context. The authors should also include any information they have about the cells that they cultured (passage numbers, number of population doublings before their experiments, etc.) for the purposes of reproducibility. Also potentially complicating interpretation of results is that the cell lines were cultured in different media and different concentrations of serum. 

The results section is lengthy and contains fairly long lists of enumerated proteins differentially expressed in vesicles from different cell lines, which is replicated in their tables and figures. It would be better for the text to focus on more overarching themes and concepts (like the GO terms and pathways and processes), while pointing out key genes that the authors think it is important for readers to notice. Also, in the results, the authors discuss the pairwise comparisons at length, but the most important thing to the reader is whether there is a difference in the protein content of vesicles from IPF-derived vs. non-IPF derived fibroblasts (different in LL29 and LL97A vs. CCD8Lu and CCD19Lu). According to the authors' venn diagram in Figure 3A, this includes 3 proteins identified in both LL group fibroblasts and 10 proteins identified in both CCD19 fibroblasts. It might be good for the authors to discuss these upfront, as they are likely the most interesting data to the readers. 

Minor points: 

Rather than giving the number of cells per plate, authors should state the density of cells and the area of the culture vessel. 

Authors should give concentrations for antibodies for western blot.

Centrifugation speed in 2.4.2 should be in xg not rpm, and concentration of trypsin used for digestion should be included. 

 Authors should give the paper a thorough readthrough for grammar/syntax issues and to clear up any unambiguous wording.

Author Response

In this manuscript, Velázquez et al. describe a set of experiments in which they extract vesicles from 2 healthy lung-derived and 2 IPF lung derived fibroblast lines, subject them to label free relative quantification by LC/MS/MS, and further characterize these differential protein profiles using bioinformatic methods. The paper has merit and presents some interesting data but, in my opinion, is hampered by a few major weaknesses that should be addressed.

1.- A notable weakness of this study, and of any other study that only investigates fibroblasts cultured in vitro, is the non-physiological culture conditions that fibroblasts are exposed to when cultured in vitro. In vitro, fibroblasts maintain some characteristics that they possessed in vivo, while losing others, limiting consequences that can be extrapolated solely from in vitro findings.

Answer: Thank you very much for this pertinent comment. We fully agree with what the reviewer states. Undoubtedly, the use of in vitro models has many drawbacks, such as the loss of particular features conferred by the interaction with a variety of other type of cells close by in the microenvironment, among many other biological factors. However, in vitro systems have been key tools as the first approach for defining molecular mechanisms associated to a pathophysiological condition, such as the activation of involved signaling pathways. Additionally, as a first approach, in vitro systems are also valuables to generate novel hypotheses and take them one step further in order to either accept or discard them by searching the in vitro findings in an in vivo system. As you can see, here we have identified some relevant molecules and physiological processes that represent an attractive aim to be address in further investigation. Taking in account this important suggestion of the reviewer, the revised version of our manuscript now include a paragraph pointing out this limitation that our research did not address.

The following paragraph now is included in the discussion section of the revised version of our manuscript, at page #17, lines #485 to #492:

In the present study, we identified differentially expressed proteins in EVs cargo isolated from human IPF fibroblast cell lines, that may play a key role in the progression of IPF. The results shown in this study may provide new insights into the molecular mechanisms of IPF. However, some limitations of this investigation, such as the need of additional experimental evidences that can validate the involvement of the identified proteins either in in vivo systems or in samples from patients bearing IPF, have yet to be addressed. Moreover, the association between the content of EVs cargo and IPF must be also investigated.

2.- The findings that the authors present here would be enhanced significantly if they demonstrated that the IPF-derived fibroblasts (preferably cultured under the conditions that the authors used to produce vesicles) demonstrated increased pro-fibrotic phenotypes compared to the healthy lung fibroblasts (increased collagen deposition, increased expression of pro-fibrotic genes like ACTA2, COL1A1, TGFB1, CCN2, etc., by IF, WB, RT-PCR, etc.), or other relevant phenotypes that would suggest that some IPF phenotypes in vivo are maintained in fibroblasts cultured in vitro relative to "healthy" lung fibroblasts. If the authors haven't demonstrated that these fibroblasts maintain some differential, IPF-related phenotypes, then it will be very difficult to convince the reader that the differential protein expression profiles that they find in their proteomic analyses (and that they use for downstream pathway analyses, etc.) are relevant to fibrosis in some way.

Answer: Thank you very much for this helpful focus on our results. We agree that to fully elucidate the mechanisms involved in IPF by using our data, it would be relevant to determine the status of the identified proteins through complementary methodologies such as IF, WB and RT-PCR. However, we decided to submit for publication our investigation based on two main reasons:

  1. By using only proteomics analysis form in vitro systems several researches have reported interesting data that have enriched the state of the art of different diseases. In general, omics analyses by themselves have generated a bulk of data in defining key knowledges to better understand the pathophysiology of diseases; for example, (Chan H et al., 2019; Geiger T et al., 2012; Li P et al., 2013).
  2. Due to the increment of infected people by SARS-CoV-2 virus in our country, the mobility and access to our institution has been importantly reduced and only very essential activities are allowed. So, activities and access of auxiliar personnel to our institutional labs are still restricted. Despite this fact, even though our reported data need to be validated in in vivo systems by using additional methodologies, we believe that our findings are of relevance and contribute in describing some features associated to the participation of extracellular vesicles cargo in the progression of IPF.

Additionally, as a follow-up to our investigation, we now are performing a meta-analysis and we pleasantly share you that our results show that some proteins identified in this investigation are also overexpressed in lung tissue samples from patients bearing IPF. Thank you very much for the observation and for sure, we will consider to include additional methodologies in our ongoing projects.

To partially cover the reviewer suggestion, we have included a statement in the discussion section at page #15, lines #353 to #360, as follow:

IPF is characterized by an increased agglomeration of fibroblasts bearing a profibrotic phenotype (29). These cells have been described as the main source of extracellular matrix that accumulates in fibrotic areas of the lung (4, 29). Several investigations have shown that lung fibroblasts isolated from IPF patients and cultured in vitro, conserve a phenotypic profile that maintain a high capability to express markers such as smooth muscle alpha actin (α-SMA), extracellular matrix-associated proteins collagen I or fibronectin, as well as, a higher proliferative capability, compared to fibroblasts isolated from normal lungs (29-31).

References (only for review)

Chan H, Bhide KP, Vaidyam A, Hedrick V, Sobreira TJP, Sors TG, et al. Proteomic Analysis of 3T3-L1 Adipocytes Treated with Insulin and TNF-α. Proteomes. 2019;7(4).

Geiger T, Wehner A, Schaab C, Cox J, Mann M. Comparative proteomic analysis of eleven common cell lines reveals ubiquitous but varying expression of most proteins. Molecular & cellular proteomics : MCP. 2012;11(3):M111.014050.

Li P, Lai X, Witzmann FA, Blazer-Yost BL. Bioinformatic Analysis of Differential Protein Expression in Calu-3 Cells Exposed to Carbon Nanotubes. Proteomes. 2013;1(3):219-39.

3.- This weakness is also exacerbated by the differences in culture medium and FBS concentration that the authors describe in the methods for culture of IPF vs. non-IPF fibroblasts.

Answer: Thank you very much for this meticulous observation. We apologize for this important mistake. We now have corrected the description in material and methods section according to what we actually did for our in vitro experiments. So, all four cell lines were maintained in the same Eagle's minimal essential medium (EMEM) supplemented with 10% FBS serum. We believe that the corrected version of our manuscript is better explained.

The following highlighted description now is included in material and methods section of the revised version of our manuscript, at page #2, lines #83 to #91:

The fibroblast cell lines LL97A and LL29, bearing a human IPF phenotype, were purchased from the American Type Culture Collection (ATCC, Cat. No. CCL-134 and CCL-191, respectively; Manassas, VA, USA). The fibroblast cell lines CCD8Lu and CCD19Lu, used as normal human lung fibroblasts, were purchased from the American Type Culture Collection (ATCC, Cat. No. CCL-201 CCL-210, respectively; Manassas, VA, USA), All cell lines were cultured in Eagle's minimal essential medium (EMEM) (ATCC No. 30-2003) supplemented with 10% SFB and 100 U/ml penicillin/streptomycin (GIBCO). Cells were maintained in a humidified incubation at 37°C and 5% CO2, and experiments were performed using passages from 8 to 12.

4.- The weaknesses of conclusions solely from in vitro study should be stated and expanded upon in the discussion section by the authors, in order to put their study in greater context.

Answer: Thank you very much for this constructive observation. In the revised version of our manuscript, we have now pointed out the limitations that imply our in vitro study.

The corrected paragraph has been included in the discussion section, at page #17, lines #485 to #492, as follow:

In the present study, we identified differentially expressed proteins in EVs cargo isolated from human IPF fibroblast cell lines, that may play a key role in the progression of IPF. The results shown in this study may provide new insights into the molecular mechanisms of IPF. However, some limitations of this investigation, such as the need of additional experimental evidences that can validate the involvement of the identified proteins either in in vivo systems or in samples from patients bearing IPF, have yet to be addressed. Moreover, the association between the content of EVs cargo and IPF must be also investigated.

5.- The authors should also include any information they have about the cells that they cultured (passage numbers, number of population doublings before their experiments, etc.) for the purposes of reproducibility.

Answer: We appreciate this particular suggestion. We have now added the required information about the cell passages number that were considered for the experiments. The required information has been included in material and methods section, at page #2, lines #83 to #91, as follow:

The fibroblast cell lines LL97A and LL29, bearing a human IPF phenotype, were purchased from the American Type Culture Collection (ATCC, Cat. No. CCL-134 and CCL-191, respectively; Manassas, VA, USA). The fibroblast cell lines CCD8Lu and CCD19Lu, used as normal human lung fibroblasts, were purchased from the American Type Culture Collection (ATCC, Cat. No. CCL-201 CCL-210, respectively; Manassas, VA, USA), All cell lines were cultured in Eagle's minimal essential medium (EMEM) (ATCC No. 30-2003) supplemented with 10% SFB and 100 U/ml penicillin/streptomycin (GIBCO). Cells were maintained in a humidified incubation at 37°C and 5% CO2, and experiments were performed using passages from 8 to 12.

Regarding the number of population doublings before their experiments, we do not have the data in terms of number; but instead, we have included the data in terms of percentage. The corrected information was included in materials and methods section “2.2. Isolation of extracellular vesicles and protein extraction”, at page #2-3, lines #93 to #108; as follow:

EVs were obtained from culture supernatants of CCD8Lu (NL-1), CCD19Lu (NL-2), LL97A (IPF-1) and LL29 (IPF-2) cell lines using ultracentrifugation. Briefly, cell lines were grown in 10 cm culture plates to 70% confluence. Them, cells were washed three times with PBS before adding fresh culture medium supplemented with 5% SFB EXO-DEPLETED to reduce contamination of EVs derived from SFB and other nanoparticles that may interfere the further analysis. After incubation for 48h (85-95% confluence), culture supernatants were collected and differentially centrifuged at 2000 x g for 30 min at 4°C to remove cell debris, followed by 12,000 x g for 30 min at 4°C to remove cell debris and apoptotic bodies. Supernatants were recovered and centrifuged at 100,000 x g for 3 h at 4°C in a TH641 rotor; them, EVs pellets were resuspended in PBS, carefully washed, and centrifuged at 100,000 x g for 3 h at 4°C. EVs were resuspended in CHAPS lysis buffer for protein extraction, and protein concentration was measured by Bradford´s method. Finally, to validate that EVs were selectively isolate specific EVs markers were detected in isolated proteins by western blot. In addition, for label-free quantitative proteomic analysis, isolated EVs were resuspended in 8 M urea lysis buffer. Protein concentration was measured by bicinchoninic acid (BCA) assay (Thermo Fisher Scientific).

6.- Also potentially complicating interpretation of results is that the cell lines were cultured in different media and different concentrations of serum.

Answer: Thank you very much for this observation. As explained in the third suggestion, we apologize for this important mistake. We now have corrected the description in material and methods section according to what we actually did for our in vitro experiments. So, all four cell lines were maintained in the same Eagle's minimal essential medium (EMEM) supplemented with 10% FBS serum. We believe that the corrected version of our manuscript is better explained.

The following highlighted description now is included in material and methods section of the revised version of our manuscript, at page #2, lines #83 to #91:

The fibroblast cell lines LL97A and LL29, bearing a human IPF phenotype, were purchased from the American Type Culture Collection (ATCC, Cat. No. CCL-134 and CCL-191, respectively; Manassas, VA, USA). The fibroblast cell lines CCD8Lu and CCD19Lu, used as normal human lung fibroblasts, were purchased from the American Type Culture Collection (ATCC, Cat. No. CCL-201 CCL-210, respectively; Manassas, VA, USA), All cell lines were cultured in Eagle's minimal essential medium (EMEM) (ATCC No. 30-2003) supplemented with 10% SFB and 100 U/ml penicillin/streptomycin (GIBCO). Cells were maintained in a humidified incubation at 37°C and 5% CO2, and experiments were performed using passages from 8 to 12.

7.- The results section is lengthy and contains fairly long lists of enumerated proteins differentially expressed in vesicles from different cell lines, which is replicated in their tables and figures. It would be better for the text to focus on more overarching themes and concepts (like the GO terms and pathways and processes), while pointing out key genes that the authors think it is important for readers to notice. Also, in the results, the authors discuss the pairwise comparisons at length, but the most important thing to the reader is whether there is a difference in the protein content of vesicles from IPF-derived vs. non-IPF derived fibroblasts (different in LL29 and LL97A vs. CCD8Lu and CCD19Lu).

Answer: Thank you so much for this relevant suggestion. In the revised version of our manuscript, we have now removed the unnecessary list of differentially expressed proteins of the different cell lines as described in section “3.3. Identification of differentially expressed proteins”, to avoid repetitive information shown in supplementary tables. In addition, the content of section “3.4. Functional ontology classification and pathway enrichment analysis of differentially expressed proteins”, was refocus in order to highlight the most important proteins associated to GO terms and KEGG pathways. We have also corrected the text in order to make it clearer and easily understandable to readers.

The following highlighted description now is included in results section of the revised version of our manuscript, at page #7-8, lines #241 to #276 and page #9-10, lines #285 to #311:

3.3. Identification of differentially expressed proteins

LFQ intensity expression values and MS/MS spectral counts were used for classifying the 331 proteins into the four EVs groups. Proteins were considered specifics of a group only when the LFQ intensity value was different from zero, and MS/MS was either equal to 2 or higher, at least in one out of two biological replicates. Venn diagram shows both the common and specific proteins present in the proteome of each EVs sample derived from different cell lines. (Figure 3A). Two hundred and sixty-two proteins were identified in NL-1, 292 in NL-2, 273 in IPF-1, and 274 in IPF-2; of which, 215 proteins were common in all EVs groups. The number of specific proteins was 9 for NL-1, 11 for NL-2, 10 for IPF-1, and 6 for IPF-2 (Table S3).

Three proteins were identified exclusively from IPF-2 and IPF-1 groups, namely lysyl oxidase homolog 1 (LOXL1), collagen and calcium-binding EGF domain-containing protein 1 (CCBE1), and fibulin-2 (FBLN2). In addition, results that showed 10 proteins were exclusively for NL-1 and NL-2 groups, namely collagen alpha-1(VIII) chain (COL8A1), prostaglandin F2 receptor negative regulator (PTGFRN), complement component C8 beta chain (C8B), basigin (BSG), integrin alpha-5 (ITGA5), dihydropyrimidinase-related protein (DPYSL2), integrin alpha-1 (ITGA1), dihydropyrimidinase (DPYS), CD109 antigen (CD109), and SH3 domain-binding glutamic acid-rich-like protein 3 (SH3BGRL3).

LFQ intensity values were used to calculate the number of differentially expressed proteins. Protein FC was calculated by subtracting the average of log2 values [Δlog2 (LFQ intensity)] between the proteins identified from IPF-1 and IPF-2, vs proteins identified from NL-1 and NL-2, and the comparison was performed in four pairwise groups; i.e., IPF-1 vs NL-1, IPF-1 vs NL-2, IPF-2 vs NL-1, IPF-2 vs NL-2). Proteins showing a FC ≥ 1 but ≤ -1 and p-value < 0.05, were considered significant.

Figure 3B shows differentially expressed proteins in the four comparisons. Results indicate that 7 proteins from IPF-1 and 38 proteins from IPF-2 (≥ 1 and p < 0.05) were up-regulated; as well as, 10 proteins from IPF-1 and 20 proteins from IPF-2 were down-regulated (≤ -1 and p < 0.05), as compared to proteome of NL-1 (Table S4, Table S6). Moreover, results also show that 5 proteins from IPF-1 were up-regulated and 15 proteins from IPF-2 (≥ 1 and p < 0.05); as well as, 14 proteins from IPF-1 were down-regulated and 13 proteins from IPF-2 (≤ -1 and p < 0.05), as compared to proteome of NL-2 (Table S5, Table S7).

Volcano plots show differentially expressed proteins between IPF-1 vs NL-1 (Figure 3C), IPF-1 vs NL-2 (Figure 3D), IPF-2 vs NL-1 (Figure 3E), and IPF-2 vs NL-2 (Figure 3F). Additionally, differentially expressed proteins were clustered and visualized in a heat map showing a consistent expression pattern and clustering between groups (Figure S1). Differentially expressed proteins identified in two or more comparison groups were 22, of which, 13 were up-regulated and 9 were down-regulated (Figure S2, Table S8).

3.4. Functional ontology classification and pathway enrichment analysis of differentially expressed proteins

Proteins identified as expressed differentially from IPF-1 and IPF-2, vs those in NL-1 and NL-2, were included for GO annotation analysis. The analysis revealed that proteins associated with IPF were mainly involved in BP, MF, and CC processes. Proteins associated to BP, such as tenascin-c (TNC), insulin-like growth factor-binding proteins 7 (IGFBP7), and fibrillin-1 (FBN1) were up-regulated and enriched for extracellular matrix organization and cell adhesion (Figure 4-A). Among down-regulated proteins, annexin A2 (ANXA2), annexin A1 (ANXA1) and aminopeptidase N (ANPEP) were associated to neutrophil degranulation and drug response (Figure 4-A).

From proteins associated to MF process, up-regulated proteins were mainly involved in extracellular matrix structural constituent and proteins binding, including TNC, IGFBP7 and FBN1 (Figure 4-B). On the other hand, down-regulated proteins were mainly involved in protein binding and identical proteins binding, including collagen alpha-1(XVIII) chain (COL18A1), annexin A5 (ANXA5) and 5-nucleotidase (NT5E) (Figure 4-B).

From proteins associated to CC, up-regulated proteins were mainly involved in the collagen-containing extracellular matrix and the extracellular region, including TNC, IGFBP7 and FBN1 (Figure 4-C). Down-regulated proteins were mainly involved in extracellular exosome and extracellular region, including COL18A1, ANXA5 and NT5E (Figure 4-C). The full list of GO analyses can be found in Table S9, Table S10, Table S11, Table S12, Table S13, and Table S14.

In addition, KEGG pathway enrichment analysis showed that up-regulated proteins were involved in receptor-EMC interaction, focal adhesion, and PI3K-Akt signaling pathways. Proteins related to these pathways were laminin subunit beta-2 (LAMB2), laminin subunit alpha-2 (LAMA2), laminin subunit alpha-5 (LAMA5) and TNC (Figure 5-A). Down-regulated proteins were mainly enriched in focal adhesion, PI3K-Akt signaling, protein adsorption, and protein digestion pathways. Proteins related to these pathways were COL18A1, filamin-C (FLNC) and integrin alpha-2 (ITGA2) (Figure 5-A). The full list of KEGG pathway enrichment analysis can be found in Table S15 and Table S16.

8.- According to the authors' venn diagram in Figure 3A, this includes 3 proteins identified in both LL group fibroblasts and 10 proteins identified in both CCD19 fibroblasts. It might be good for the authors to discuss these upfront, as they are likely the most interesting data to the readers. 

Answer: Thank you so much for the support to improve our manuscript. We apologize the omission of this relevant finding. So, we have now included a description of this finding in the results section “3.3 Identification of differentially expressed proteins”, at page #8, lines 250 to #257; as follow:

Three proteins were identified exclusively from IPF-2 and IPF-1 groups, namely lysyl oxidase homolog 1 (LOXL1), collagen and calcium-binding EGF domain-containing protein 1 (CCBE1), and fibulin-2 (FBLN2). In addition, results that showed 10 proteins were exclusively for NL-1 and NL-2 groups, namely collagen alpha-1(VIII) chain (COL8A1), prostaglandin F2 receptor negative regulator (PTGFRN), complement component C8 beta chain (C8B), basigin (BSG), integrin alpha-5 (ITGA5), dihydropyrimidinase-related protein (DPYSL2), integrin alpha-1 (ITGA1), dihydropyrimidinase (DPYS), CD109 antigen (CD109), and SH3 domain-binding glutamic acid-rich-like protein 3 (SH3BGRL3).

In addition, we have extensively discussed these important results in the discussion section, specifically at page #15-16, lines #384 to #423; as follow:

It is important to point out that some of proteins identified as exclusive of IPF-1 and IPF-2 groups have been studied because of their close relationship with the IPF pathogenesis. For example, LOXL1 an enzyme that belong to lysyl oxidase (LOX) family and that catalyze the cross-linking between collagen and elastin by promoting the oxidative deamination of lysine residues and contributing to the stiffness of ECM in different tissues (32-34). Moreover, it has been shown that the main LOXL1-producing cells are fibroblasts and myofibroblasts in fibrotic lung, and this enzyme has been mainly localized into dense fibrosis and fibroblastic foci (33, 34). Increased LOX1 levels have been associated with increased proliferation, invasion, resistance to apoptosis and shrinkage of fibroblasts (33, 34). Studies in murine models of IPF have demonstrated that LOXL1 -/- mice are protected from IPF induced by AdTGF-β1 administration (32, 34). However, the role of LOXL1 in the pathogenesis of IPF has not been fully described.

FBLN2 is a glycoprotein belonging to the fibulin family, which is predominantly localized in the basement membrane of extracellular matrix and elastic basal fibers (35). It has been described that FBLN2 can regulate different signaling pathways associated to cell metabolism, migration, proliferation and differentiation (35). FBLN2 is also up-regulated during the tissue remodeling process, and its overexpression has been associated with different chronic diseases such as fibrosis and cancer (35, 36). In vitro studies performed in cardiac fibroblasts isolated from Fbln2 -/- mice showed that FBLN2 plays an essential role in angiotensin II-induced TGF-β signaling pathway, while studies performed in a murine model of cardiac hypertrophy demonstrated that Fbln2 -/- mice show attenuated development of cardiac hypertrophy induced by chronic infusion of subpressor and pressor doses of angiotensin II (36). Additionally, the role of FBLN2 has been also studied in a murine model of myocardial fibrosis induced by chronic infusion of pressor doses of angiotensin II showing that the fibrosis degree decreases in Fbln2 -/- mice (37). However, its participation in the pathogenesis of IPF had not been described.

The involvement of CCBE1 in IPF has not been described. However, CCBE1 has been reported to be associated with extracellular matrix remodeling, development of multicellular organisms and development of lymphatic vessels (38, 39). Currently, the relationship of CCBE1 with different types of cancer has been studied (39). However, data indicate that CCBE1 may have particular effects depending on the type of cancer, which has generated some controversy about its role in cancer development (38). An investigation showed that in patients bearing lung cancer, CCBE1 expression is down-regulated within tumor tissue which is associated with poor prognosis (39). In contrast, CCBE1 is overexpressed in colorectal cancer tissue compared to adjacent tissues, Results showed that CCBE1 overexpression significantly correlates with increased lymph node metastasis, vascular invasion, and liver metastasis (38). Thus, our result indicates that CCBE1 might be also participating in the IPF progression and suggests that it could be contributing in the fibrogenesis associated to cancer, such as the fibrosis that precede the liver cancer appearance, an intriguingly phenomenon that should be addressed.

Minor points:

1.- Rather than giving the number of cells per plate, authors should state the density of cells and the area of the culture vessel. 

Answer: We agree with this important suggestion. In this regard, section “2.2. isolation of extracellular vesicles and protein extraction”, was corrected to delete the previous confusion. We have now included the percentage of cells per plate and the diameter of the culture plate, at page #2-3, lines #93 to #108; as follow:

EVs were obtained from culture supernatants of CCD8Lu (NL-1), CCD19Lu (NL-2), LL97A (IPF-1) and LL29 (IPF-2) cell lines using ultracentrifugation. Briefly, cell lines were grown in 10 cm culture plates to 70% confluence. Them, cells were washed three times with PBS before adding fresh culture medium supplemented with 5% SFB EXO-DEPLETED to reduce contamination of EVs derived from SFB and other nanoparticles that may interfere the further analysis. After incubation for 48h (85-95 % confluence), culture supernatants were collected and differentially centrifuged at 2000 x g for 30 min at 4°C to remove cell debris, followed by 12,000 x g for 30 min at 4°C to remove cell debris and apoptotic bodies. Supernatants were recovered and centrifuged at 100,000 x g for 3 h at 4°C in a TH641 rotor; them, EVs pellets were resuspended in PBS, carefully washed, and centrifuged at 100,000 x g for 3 h at 4°C. EVs were resuspended in CHAPS lysis buffer for protein extraction, and protein concentration was measured by Bradford´s method. Finally, to validate that EVs were selectively isolate specific EVs markers were detected in isolated proteins by western blot. In addition, for label-free quantitative proteomic analysis, isolated EVs were resuspended in 8 M urea lysis buffer. Protein concentration was measured by bicinchoninic acid (BCA) assay (Thermo Fisher Scientific).

2.- Authors should give concentrations for antibodies for western blot.

Answer: We sincerely appreciate this observation and apologize for the omission. The antibody concentration was indicated in section “2.3 Western Blot” of materials and methods, at page #3, lines #110 to #118; as follow:

Western blot (WB) analysis was performed according to standard protocols. Briefly, 50 µg of total protein isolated from EVs were separated by SDS-PAGE gel electrophoresis. Then, proteins were transferred to a PVDF membrane and primary antibodies Anti-HSP90-αβ (1:500; SC13119; Santa Cruz biotechnology), Anti-Alix (1:500; SC53540; Santa Cruz Biotechnology), and Anti-Flot-1 (1:500; BD 610821; BD biosciences) were incubated overnight at 4°c. Protein expression was visualized after membranes were incubated with horseradish peroxidase-conjugated anti-mouse secondary antibodies (1:5000; SC-516102; Santa Cruz Biotechology) and then, protein spots were revealed by using 1-Step Ultra TMB-Blotting reagent (Thermo Fisher Scientific).

3.- Centrifugation speed in 2.4.2 should be in xg not rpm, and concentration of trypsin used for digestion should be included. 

Answer: Thank you so much for this complementary suggestion. In the revised version of our manuscript, we have now converted the centrifugation speed units and added the trypsin concentration in section “2.4.2 Sample preparation, at page #3, lines #128 to #139, as follow:

First, total protein lysates from EVs samples were centrifuged at 16,000 x g, 4°C for 15 min, and supernatants were transferred to a clean Eppendorf tube. Then, total proteins were precipitated from protein solution by using cold acetone and centrifuged at 16,000 x g, 4°C for 15 min, and supernatant was discarded. Subsequently, protein pellets were dissolved in 6 M of aqueous urea solution, and 30 μg of total protein was denatured with 10 mM DTT by incubating it at 56°C for 1 h, followed by alkylation with 50 mM IAA and incubated at room temperature for 60 min in the dark. Next, 500 mM ammonium bicarbonate was added to the solution to get a final concentration of 50 mM ammonium bicarbonate at pH 7.8. Then, trypsin (Promega) was added to protein solution at 1:50 ratio for digestion at 37 °C for 15 h. Digested peptides were further purified with a zip tip to remove the salt. Finally, samples were dried under vacuum and stored at -20 °C, for further analysis.

4.- Authors should give the paper a thorough readthrough for grammar/syntax issues and to clear up any unambiguous wording.

Answer: We really appreciate the reviewer suggestion. By taking into account this important comment, a native English speaker has completely reviewed the grammar and syntax of our manuscript. We believe that the revised version of our manuscript has now substantially improved.

Additionally, in order to make reading more fluent, we have shortened the group nomenclatures and now they are named as follow: for LL97A, LL29, CCD8Lu and CCD19Lu cell lines, names now appear as IPF-1, IPF-2, NL-1 and NL-2, respectively, throughout the manuscript, including figures, figures legends, tables and supplementary materials.

Finally, we are very thankful with the pertinent suggestion/comments of the Reviewer #1, and we hope that changes in the revised version of our manuscript are satisfactory and acceptable for publication by Biomedicines journal.

Reviewer 2 Report

The authors investigated extracellular vesiscles in idiopathic pulmonary fibrosis through proteomic approach. The manuscript is original and few literature data is available about the investigation of EVs in IPF. The statistical analysis was appropriate and complete. Figures and tables were appropriate. I suggest to accept the manuscript in the present form.

Author Response

The authors investigated extracellular vesiscles in idiopathic pulmonary fibrosis through proteomic approach. The manuscript is original and few literature data is available about the investigation of EVs in IPF. The statistical analysis was appropriate and complete. Figures and tables were appropriate. I suggest to accept the manuscript in the present form.

Answer: Thank you very much for the encouraging comment about our investigation. We hope that our report contributes in the underlying molecular mechanisms associated to IPF pathogenesis.

Thank you very much for helping us to improve our manuscript!

Round 2

Reviewer 1 Report

With their inclusion of study limitations, the authors have addressed my concerns adequately.